# Implementation Models of Compassionate Communities and Compassionate Cities at the End of Life: A Systematic Review

**DOI:** 10.3390/ijerph17176271

**Published:** 2020-08-28

**Authors:** Silvia Librada-Flores, María Nabal-Vicuña, Diana Forero-Vega, Ingrid Muñoz-Mayorga, María Dolores Guerra-Martín

**Affiliations:** 1New Health Foundation, 41004 Sevilla, Spain; silvia.librada@newhealthfoundation.org (S.L.-F.); ixmayorga@gmail.com (I.M.-M.); 2Palliative Care Team, Arnau de Villanova Hospital, 25198 Lleida, Spain; mnabalv@gmail.com (M.N.-V.); dforerovega@hotmail.com (D.F.-V.); 3Department of Nursing, University of Sevilla, 41009 Sevilla, Spain

**Keywords:** community networks, empathy, palliative care, review

## Abstract

In the last decade, we have seen a growth of Compassionate Communities and Cities (CCC) at the end of life. There has been an evolution of organizations that help construct Community-Based Palliative Care programs. The objective is to analyze the implementation, methodology and effectiveness of the CCC models at the end of life. We conducted a systematic review following PRISMA ScR Guideline. The protocol was registered on PROSPERO (CRD42017068501). Five databases (MEDLINE, EMBASE, Web of Science, CINAHL and Google Scholar) were searched for studies (from 2000 to 2018) using set eligibility criteria. Three reviewers screened full-texts articles and extracted study data. Outcomes were filled in a registration form which included a narrative synthesis of each article. We screened 1975 records. We retrieved 112 articles and included 31 articles for the final analysis: 17 descriptive studies, 4 interventions studies, 4 reviews and 6 qualitative studies. A total of 11 studies regard the development models of CCC at the end of life, 15 studies were about evaluation of compassionate communities’ programs and 5 studies were about protocols for the development of CCC programs. There is poor evidence of the implementation and evaluation models of CCC at the end of life. There is little and low-/very low-quality evidence about CCC development and assessment models. We found no data published on care intervention in advance disease and end of life. A global model for the development and evaluation of CCC at the end of life seems to be necessary.

## 1. Introduction

In the last decade, there has been a growing interest in the development of Compassionate Communities and Compassionate Cities (CCC) at the end of life. This movement is based on the World Health Organization’s Ottawa Charter [1] and promotes the motivation of communities to take more responsibility in their healthcare improving the care of people at the end of life [2]. From this public health approach three main models can be distinguished: Models from the health services to the community (top-down approach).Models from community participation through the development of actions and events that involve communities in the promotion of their health (bottom-up approach).Models from organizations and community participation that ensure that population needs and desires are covered by the community’s impulse and offer tools and techniques to assess these needs and propose solutions.

CCC movement at the end of life was consolidated with the definition of Compassionate City Charter promoted by the Public Health and Palliative Care International (PHPCI Organization) that defines “Compassionate Cities as those that publicly recognize people at the end of life and their needs and are aware of the search and involvement of all the main sectors of the city to help through care and accompaniment to reduce the social, psychological and health impact of life’s difficult processes and situations, especially those related to disability, ageing, dependence, end of life, burden of caregivers, pain and loss of a loved one” [3].

A Compassionate City [3]:Has local health policies that recognize compassion as an ethical imperative.Meets the special needs of elders, those living with life-threatening illnesses, and those living with loss.Has a strong commitment to social and cultural differences.Involves grief and palliative care services in local government policy and planning.Offers its inhabitants access to a wider variety of supportive experiences, interactions and communication.Promotes and celebrates reconciliation with indigenous peoples and memory of other important community losses.Provides easy access to grief and palliative care services.

Through the Compassionate Communities approach:Death, dying and bereavement would cease to be taboo subjects and would become more normalized within society.People’s expectations of death and dying will change, as well as how death will be managed.Palliative care will be re-oriented, supporting health and social care staff to work with the community in providing care to those at the end of life and their loved ones.

In the last years, several organizations from different countries (United Kingdom [4], Scotland [5], Ireland [6], Austria [7], India [8], Canada [9] Australia [10], Colombia [11] and Spain [12,13]) have boosted the development of Compassionate Communities and Cities with different models. 

These experiences, centered on Community-Based Palliative Care Models, provide an opportunity for palliative care to progress toward a new vision linked to public health and integrated care models.

At the end of life, compassion—defined as the ability to identify and understand the suffering of another person and the desire to alleviate it—also plays an important role for patients, families, networks of care and supportive care providers. It is part of the Palliative Care definition. Harnessing the power of compassion to aid those dying and support their loved ones could provide invaluable information on how to promote optimal levels of healthcare throughout the entire lifespan. In a recent review about compassion in palliative care [14], an improvement in healthcare outcomes has been demonstrated related to the quadruple aim: improvement of patient’s experience, population health, self-care of professionals, healthcare provider’s satisfaction and reduction of healthcare costs. This evidence has motivated the design, development and nurturing of Compassionate Communities and Cities. This can enhance our collective ability to care for each other at the end of life [15].

Despite this incipient movement, we have not found a systematic framework for the development of Compassionate Communities programs. There are no validated tools to measure the effectiveness of these programs and their impact on patients and their families’ well-being or the health systems. 

We have reviewed the existing evidence related to the implementation models of Compassionate Communities and Cities at the end of life to identify their methodology and effectiveness assessment system. This systematic review was undertaken to answer the following research questions:How many end of life Compassionate Communities and Cities development models exist?What are their methods, processes and measures to allow the intervention assessment?Could we compare different degrees of development of Compassionate Communities and Cities in different countries and organizations?

## 2. Materials and Methods 

### 2.1. Design 

We followed PRISMA-ScR Guidelines for review. The review protocol was registered on PROSPERO (CRD42017068501) before screening and data extraction (date of publication July 2017) available on https://www.crd.york.ac.uk/PROSPERO/display_record.php?ID=CRD42017068501&ID=CRD42017068501.

### 2.2. Search Strategy

The electronic databases used were: MEDLINE (PubMed), EMBASE, Web of Science, CINAHL and Google Scholar. The search strategy was based on MEDLINE search Mesh terms and adapted for other databases if it was necessary (Table 1). Other databased as DARE, SUMSEARCH and Cochrane and institutional resources were used. Grey literature as conference abstracts, books and reports were also consulted. The search was limited to studies published from 2000 to 2018. 

### 2.3. Study Eligibility Criteria

Table 2 shows the predefined inclusion criteria used to select eligible studies. 

The predefined criteria exclusion used to select eligible studies are: 1. Articles or other types of documents incomplete or in the process of being drawn up; 2. Documents that do not include information related to the object of the search.

### 2.4. Type of Studies 

We included both qualitative and quantitative studies reporting original data: descriptive, retrospective and/or prospective studies, intervention studies and systematic reviews. 

### 2.5. Quality Assessment

All studies were assessed for methodological quality by the Grading of Recommendations, Assessment, Development and Evaluation system (GRADE). One reviewer (S.L.-F.) graded all the studies and this was verified by a second reviewer (M.N.-V.). 

### 2.6. Screening and Data Extraction

Searches were conducted by one author (S.L.-F.). The documentation was selected by the appearance of the keywords in the reports and articles’ titles and abstracts. All references were extracted in Microsoft Excel panel regarding the title, author, journal, data, full text available, type of document, type of study, and information contained.

To reduce the selection risk of bias, the first identification was made by title and abstract, trying to select those documents that answer the following questions:Does the article discuss Compassionate Communities and Cities at the end of life?Do they regard the development model of Compassionate Communities and Cities?Do they show the phases of development or standards?Are the development experiences and results of Compassionate Communities and Cities discussed?Is it a pilot study?Is the development method described?Do they describe tools and resources used to develop Compassionate Communities and Cities?Was it developed from public policies in palliative care?

The full-text papers (n = 42) were review by three authors (D.F.-V., I.M.-M. and S.L.-F.) in a peer review process in order to finally select the required documentation. A registration form was designed to extract the information from articles and by a checklist and a description of the most relevant results of the papers reviewed (Table 3). Finally, all authors participated in the discussion and synthesis of all the article selected. Selected articles were reviewed by two experts in systematic reviews (M.D.G-M. and M.N.-V.). 

### 2.7. Data Analysis

Each author completed the registration form independently. This included a narrative synthesis of each article describing: aims, study design, participants/scope, country and main findings. The most important outcomes were filled in the registration form according to the following criteria:Objectives: What do they want to achieve at city, organization, country level?Scope: What is the scope or coverage?Development models How do they do it?Degree of development: pilot, initial or consolidatedStandards: Are there goals to be achieved? What are the standards to achieve them?Tools, resources, evaluation models and analysis of results.

Disagreements among reviewers were resolved through discussion among all authors to obtain the definitive information. 

## 3. Results

A total of 1975 articles were identified by databases searching. A total of 1863 were excluded because they were duplicated (n = 720) or because they did not match eligibility criteria (n = 1143). A total of 112 articles were retrieved but 70 articles were excluded by title and abstract because they were considered not related to CCC models by the three reviewers. This resulted in 42 full-text articles. A total of 11 articles were excluded by reviewers in a discussion session because they did not offer relevant information. We obtained 31 articles for the final analysis. Figure 1 represents the selection process.

Appendix A summarizes the main characteristics and the quality assessment of the included studies: 17 descriptive studies, 4 interventions studies, 4 reviews and 6 qualitative studies published from 2000 to 2018. Evidence levels have been matched by 25 low-level and 6 very low-level articles. 

Data were extracted into a data extraction table with the most relevant information (Appendix A). The results have been classified into three topics according to the objectives of the review (key themes).

### 3.1. Compassionate Community or Cities at the End of Life Development Models (Theme 1)

Eleven documents offered information about CCC development models. Ten were descriptive studies and one was a systematic review.

Regarding public policies, we found organizations and governments, such as Canada and U.S, that are making important efforts to offer an attention more integrated, compassionate, effective and less expensive at the end of life. The U.S Healthcare System have published seven recommendations that served as a model for the development of compassionate communities [16]: “Commitment to compassionate healthcare leadership, to teach compassion, to value and reward compassion, to support caregivers, to partner with patients and families, to build compassion into healthcare delivery and to deepening our understanding of compassion.”

Abel et al. [17] described the UK National Health Council approach based on public health and community principles too. This council included key elements for the development of Compassionate Communities, such as education for death in schools and workplaces, decision making of people about the end of their life, availability of community resources including social volunteering, favoring place of care, death at home and greater involvement of the community in end of life care. Paul and Sallnow [4] and De Zulueta [18] studies considered community promotion actions a priority with a greater impact on schools, promoting community involvement through social awareness, compassionate leadership and networking.

Models of Compassionate Communities described by Chou et al. [19] and Sallnow et al. [20] add valuable elements as the development of coalitions that bring together professionals, community networks, organizations and researchers to improve end of life patients’ quality of life. These coalitions provided a space to share experiences and to create social networks around these people. From this perspective, Blinderman [21] urged that palliative care must be considered as a right and be offered in a dignified and compassionate way even in low-income countries where they are not yet well implemented.

Care, compassion and community have been highlighted by Kayser et al. [22] in a descriptive study with 33 end of life patients’ care needs as a triple element to guarantee the adequate attention of people at the end of life. 

Regarding the creation and management of care networks around people at the end of life, the model published by Abel et al. [23] named Circle of care describes that patients must be surrounded by their family and close friends, from the community and service providers.

This “circle of care model” is applied as a tool to treat the person and their entire community as parts of the network [23].

Sallnow et al. [20] describe a neighborhood network in palliative care that involves a trained community to respond to end of life patients’ needs: social, spiritual and emotional needs, as well as, the proper management and control of symptoms in a domiciliary context.

Flager and Dong [24] also provides evidence that home care reduces hospitalizations and visits to emergency departments; thus, encouraging care and accompaniment in this environment can lead to more cost-efficient programs.

Kellehear [25] argues that end of life care is everyone’s responsibility, and that compassionate cities and communities are an example of health responsibility. Both people and organizations have a fundamental role in their development.

### 3.2. Evaluation Models of Compassionate Communities and Cities: Indicators, Standards or Data That Allow Evaluating the Organizations and Resources (Theme 2)

We have identified 15 documents with information about indicators, standards and partial or global evaluation models for CCC: 3 descriptive studies, 3 evaluations carried out through intervention processes, 6 qualitative studies and 3 systematic reviews. 

The results of this mater have been grouped into three blocks: Compassionate Communities and Cities evaluation models/ systems.Evaluation of specific Compassionate Communities and Cities programs.Evaluation of compassion practice.

#### 3.2.1. Compassionate Communities and Cities Evaluation Models/Systems

We found two reviews on CCC evaluation models.

Sallnow et al. [26] offer a meta-ethnographic evaluation system to explore experiences of patients, caregivers and professionals. These experiences evaluated: caregiver’s satisfaction, professional and patients’ experiences, and “the community development since the creation of networks of care and educational programs”. The quantitative findings mapped to the meta-ethnography can lead to improved outcomes for carers, such as decreased fatigue or isolation and increase in size of caring networks. These wider social networks can influence place of death and palliative care services involvement.

Pfaff and Markaki [27], in an integrative review of the palliative care and end of life literature, selected several indicators and attributes for the evaluation of Compassionate Communities related to: Overarching Structures: Patient and Family-Centered Care,Overarching Values: Empathy, Sharing, Respect, Partnership.Process indicators: Communication, Shared decision-making, Goal setting;Results indicators: organizational development and satisfaction.

The great contribution of this review was the development of an assessment model that could be use by governments, healthcare administration and providers for improve health, strengthen the provision of care and control health costs [27].

Regarding the evaluation of community interventions, the work of Luzinski et al. [28] analyzed cost efficiency and service quality. These authors calculated, through the case manager intervention the annual cost savings provided directly to clients as a result of community case management interventions, that the case managers saved an average of $93,000 per year for the CCM group of clients.

#### 3.2.2. Compassionate Communities and Cities Evaluation Models/ Systems

Canada’s Compassionate Care Benefit (CCB) program used a multimodal evaluation by caregivers and provider’s interviews:Caregivers were asked about aspects related to the program implementation: eligibility, information, timing and financial opinion by comparing ideal expectation with the reality [29,30,31].Providers underwent semi-structured telephone interviews on aspects related to Compassionate Care Benefit usefulness, its access facilitators and barriers; experiences related to recommending the Compassionate Care Benefit to potential applicants and suggestions for program improvement [31].

These works have not allowed us to identify quantitative assessment indicators. Nevertheless, these authors point out that they have obtained a qualitative critical analysis that help them to make decisions in four specific sub-themes: temporal, financial, informational and administrative aspects related to the program [32,33]. Findings demonstrated that participants expect the CCB to provide: (1) Temporal: an adequate length of leave time from work, which is reflective of the uncertain nature of caregiving at end of life; (2) Financial: adequate financial support; (3) Informational: information on the programme to be disseminated to front-line palliative care providers so that they may share it with others; and (4) Administrative aspects: a simple, clear, and quick application process. 

The program published by Pesut et al., N-CARE, developed their evaluation based on the opinion of seven volunteer patients living with advanced chronic illness. They explored satisfaction and areas of improvement after 12 months of accompaniment. This assessment used questionnaires and semi-structured interviews [9].

In an Irish context, Milford Care Centre [34] structured the evaluation by: The visibility of the program: web, marketing materials, brochures, awareness campaigns, etc.Citizen participation: community groups, volunteering, conversation groups, workshops, etc.The social care model: Community mentors, community partners, voluntary community mentors recruited and trained, community mentors advertised in the designated areas and with all appropriate service providers.

#### 3.2.3. Compassionate Communities and Cities Evaluation Models/Systems

Crowther et al. [35] published a study conducted to measure the effect of compassion, humanity and kindness on patients with dementia during their last year of life. This is a qualitative analysis based on interviews with 40 caregivers in their own homes or places of work, recorded and transcribed. These authors used a narrative methodology with the goal of obtaining the experiences of the patient caregivers themselves. From the methodological point of view does not contain a structured script. Results revealed that compassion increases humanity and kindness approach to each person’s pain, distress, anxiety or need. At the same time, however, it highlights the need of better education on compassion for health and social professionals to develop a high-quality compassion attention. 

Martins et al. [36] propose a scale tested on 310 professionals that can be used to promote understanding about the impact of the compassion level in the disease, in interpersonal relationship and in the community. The scale consists of 5 elements and 10 items—Generosity (4 items), Hospitality (2 items), Objectivity (2 items), Sensitivity (1 item) and Tolerance (1 item)—that are associated with the elements of compassion. This scale was validated and evaluated. It is friendly user, easy to score and characterized by good psychometric properties with an internal consistency of 0.82.

The review published by Sinclair et al. [37] to define what we understand by compassion and what components can be associated with it, offers six elements arising from 44 studies that explored perceptions of compassionate care: nature of compassion, development of compassion, interpersonal factors related to compassion, action and practical compassion, barriers and enablers of compassion, and outcomes of compassion and the associated aspects of compassion in clinical interventions and aspects of education [37]. Although this review has been able to obtain a more complete definition of compassion in healthcare, it is in the study of Sinclair et al. [38] where compassion is evaluated through a semi-structured interview model with seven key categories: virtues, relational space, virtuous response, seeking to understand, relational communicating, attending to needs and patients reported outcomes, which in turn are broken down into 27 topics and 18 sub-themes. 

From the point of view of the application of compassion, a study by Shih et al. in students [39] has allowed to evaluate several factors about improvement in Compassionate Care knowledge and factors influencing improvement in ethical decision making.

### 3.3. Compassionate Communities and Cities: Tools, Protocols or Information Systems (Theme 3)

We have identified five articles that include useful tools for the development of CC. Four of them were descriptive studies and one was an intervention study.

Dewar and Cook [40] evaluated a program of leadership and culture of compassion describing the design of a questionnaire that allows professionals to measure a series of aspects such as relationships with oneself, with patients and family and with teams and organizations. 

Moore et al. [41] described and evaluated a model of intervention in compassion with 30 nursing home residents with advanced dementia during a 6 month intervention process. They compared two nursing homes. Intervention was correctly described and analyzed by two components: (1) integrated, interdisciplinary assessment and care, and (2) education and support for paid and family carers. Data were collected in a mixed method: by semi-structured interviews and by quantitative data of cost of the interdisciplinary care leaders. Interventions improved in advance care planning, pain management and person-centered care. The 6 months of costs were £18,255. This intervention model is correctly described and has been replicated by Elliot et al. [42] in other contexts with similar population.

We found another similar intervention protocol. INSPIRE social intervention protocol by McLoughlin et al. [6] (Investigating Social and Practical suppoRts at the End of life), developed to assess the intervention process of a program called Good Neighbours Partnership (a new volunteer-led model of social and practical care/support for community underselling adults who are living with advanced life-limiting illness). It is a research protocol which randomizes patients and carers in two groups: control (standard care) and intervention (Good Neighbours Partnership) by patients and carers interviews developed before, during and after the intervention. Primary outcomes measured was the effect of the intervention on social and practical needs. The secondary outcomes were: quality of life, loneliness, social support, social capital, unscheduled health service utilization, caregiver burden, adverse impacts and satisfaction with intervention. Volunteers were also assessed in terms of their death anxiety, death self-efficacy, self-reported knowledge and confidence with eleven skills considered necessary to be effective GNP volunteers. This research protocol was developed to enroll a sample of 80 patients and their caregivers in Ireland but there are no results available. Besides, this proposal offers ideas about how to measure this kind of interventions. 

ELSA protocol by Whalse et al. [43] for the End of Life Social Action is a research proposal. This is a randomized wait-list controlled trial protocol. This project’s aim is to impact of volunteers’ initiatives to deliver befriending services to people anticipated to be in their last year of life. The project would like to include two sites in England. It will assess by validated scales: quality of life, loneliness and social support at the beginning and at week and 8.

## 4. Discussion

This review reflects the growing development of CCC that has been launched in recent years. The model described by Kellehear A [3,25] has allowed these initiatives to be oriented towards the elements that characterize the development of a Compassionate City.

Recommendations and coalitions published about the development of CCC also reflects the empowerment of this movement from public health and palliative care policies in an integrative health-social-community care model [16,17,20].

The eleven studies identified in Theme 1, “CCC development models”, describe actions from bottom up approach. They focus on social awareness, the need of training about end of life care and how to involve society for the creation of networks of care around people at the end of life. We would like to highlight the key elements for Compassionate Communities or Cities development models shared by several authors: social awareness and education programs on compassion and networks of care [4,18], programs for training caregivers, neighborhood network in palliative care [20] to provide home-based palliative care involving volunteers and the community and networks of care round people at the end of life initiatives with the implication of inner and outer networks, communities and service delivery organizations [17]. Only two of these studies included samples related to palliative care (220 palliative care providers [4] and 33 patients at the end of life [22]). By the end of this review, there were very few intervention papers on CCC components, development or assessment. 

Although these initiatives are well-described, we have not found studies that integrate global results of these processes and their application to a specific communities or cities. Viajy and Monin [44] and Herrera et al. [15] agree on the need of some key elements for proper CCC beginning and development: leadership, well defined coverage, annual work agenda, collaboration among institutions (from health, social and community areas); development of community intervention structures; community, volunteer and neighborhood’s networks activation; general population sensibilization and capacitation; assessment systems design; and mass media implication. Some of them have been already tested in a pilot program in Spain with good results [15].

Following this premise, a specific method for the development of Compassionate Communities and Cities [12] has been developed and is being extended and evaluated in several cities. These components mainly focus on the development of partnerships with the organizations (schools, companies, universities, etc.) and on the activities needed to improve awareness and train abilities to develop community networks around people at the end of life. 

Studies in this review about Theme 2 “Evaluation models of Compassionate Communities and Cities” express the benefits of compassion and community involvement in improving patient care, family and network of care satisfaction. Although satisfaction can be a good beginning, other evaluation strategies have to be developed. Satisfaction is a soft assessment tool because it is seriously influenced by expectations, the time of assessment and the memories of the responders. 

Intervention studies identified [28] have demonstrated cost effectiveness on a sample of 400 patients (average cost of $93,000 saved per year). Another Case study [45] have begun to identify and incorporate cost-effective measures of CCC programs, as the community program of Frome whose cost benefits represent 5% of the total healthcare budget (nationally, emergency admissions account for nearly 20% of the healthcare budget). 

The benefits of compassion programs can be assessed by their effects in an ethical decision-making process and ethical behaviors. An intervention performed on 251 preclinical medical students demonstrated the improvement of medical student’s competences in making more appropriate ethical decisions in end of life care [39].

The indicators identified in this review have been classified in structure, process and results and oriented to the Charter of Compassion recommendations for the development of CCC [26,27]. These allow us to confirm that it is appropriate to have a specific method in order to measure the evolution of different programs. We have not found in this review comparative studies of programs due to the lack of consensus on the measure of indicators. 

Five studies have been found for Theme 3 about the use of specific tools and protocols for the development of interventions in the community. 

Intervention models were analyzed using different measures in different settings: residents with advanced dementia [41,42]. McLoughlin et al. [6] and Walshe et al. [43] carried out an intervention protocol to promote compassionate communities in end of life patients. On the other hand, Dewar and Cook [40] carried out a leadership and culture of compassion program, after which the participants were able to detect their strengths.

These protocols have served as the basis for the development of new intervention proposals in different international contexts such as RedCuida’s protocol by Librada et al. [46] which is being implemented among communities surrounding people with advanced disease and at the end of life. 

When looking at these protocols we can observe that they are not comparable because they do not share aims and frameworks, but they do share some outcomes, such as quality of life, decrease in loneliness, increase of the number of care networks, decrease in the main carer burden that can offer only a general overview. We did not find any study comparing systematic compassion interventions vs ordinary care to assess compassion effectiveness in end of life care. Therefore, there is a lot of work to do. 

Even this review offers interesting information on recommendations and an approach to CCC models, methods and assessment systems, the quality of this evidence is low or very low, according to GRADE system. Most of them are descriptive or proposal for future interventions. We did not find any study conducted with a representative sample and randomize methodology to offer better information about the benefits of this type of interventions. More research is needed to clarify and improve our knowledge.

Regarding the limitations, this review has been based on publications of scientific articles related to models for the development of Compassionate Communities and Cities at the end of life. As these are model publications, little evidence of application studies on specific populations is detected where the development of these interventions in the community is assessed. Reports or books, norms or monographs describing development experiences have not been included. 

In addition, there is a lack of knowledge about the evolution of some of these programs. It is unknown whether they are pilot programs or are still ongoing. We have found further more descriptive studies and reviews that provide little evidence, compared to intervention studies. Most of the intervention studies have been carried out on small samples. Nevertheless, these results serve to guide models of these programs’ benefits; comparison among different initiatives developed are not comparable, due to the method used and the absence of quantitative results.

The lack of specific CCC development methods and evaluation models has not allowed us to make a comparative analysis. The need to work in this direction is reinforced.

No studies have been identified that demonstrate the opportunities or difficulties when launching projects of compassionate cities and communities. As it is an emerging movement, the experiences described should also go in this direction to guide other cities and organizations.

## 5. Conclusions

The results of this review show that there is little and low-/very low-quality evidence about CCC development and assessment models. Few data on key elements for the CCC development were found.

Evaluation methodology in the existing projects are mostly focus on local aspects. Existing evaluation models for CCC interventions are based on systematic reviews. Carers’ satisfaction and carers’ opinions have been the most common assessment approach. No data on advance disease and end of life care interventions have been yet published. 

A global model for the development and evaluation of CCC at end of life care seems to be necessary. Systemizing the processes will help emergent organizations or communities to develop Compassionate Communities and Cities and it will facilitate the assessment or its impact and effectiveness.

## Figures and Tables

**Figure 1 ijerph-17-06271-f001:**
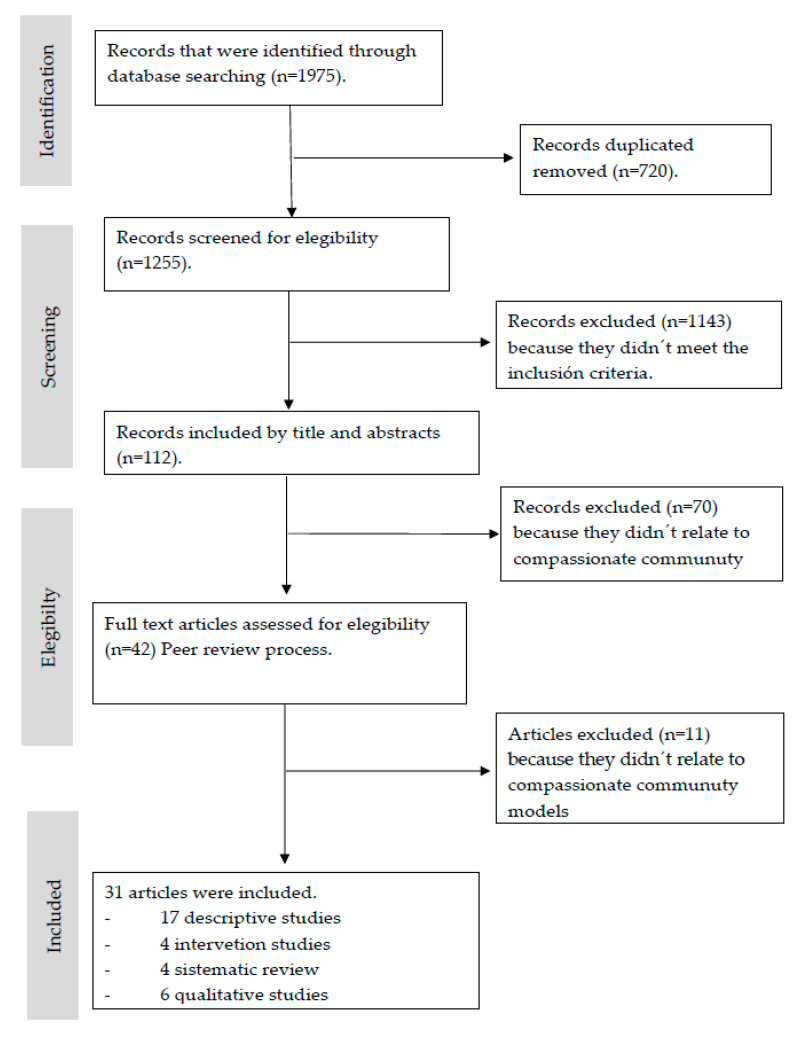
The flow diagram of literature search and selection of articles.

**Table 1 ijerph-17-06271-t001:** Search strategy.

Search Strategy
(Terminal Care OR Palliative Care OR Hospice Care OR Hospices) AND
(Community networks OR Community health planning OR Community participation OR Social supports OR Volunteers OR Social network) AND
(Governments OR policymaker OR health and social policy OR public policy) AND
(Empathy OR Compassion) AND
(Model OR Organizational OR Program development OR Standards OR Impact OR Evaluation OR measure OR outcome) AND
(Efficiency OR effectiveness OR efficacy OR impact OR excellence OR referral)

**Table 2 ijerph-17-06271-t002:** Inclusion criteria.

**Type of Documents**
National strategies, Frameworks, Strategic Plans and Reports Reports and studies evaluating the results of the development of Compassionate Communities and Compassionate Cities. Books, White papers. Tools and guides for program evaluation and measurement of indicators. Scientific articles.
**Information Contained in these Types of Documents**
Models of Compassionate Communities and Compassionate Cities at the end of life. Evaluation models of Compassionate Communities and Compassionate Cities: Indicators, standards, or data that allow to evaluate the organizations and resources. Tools, protocols or information systems that allow the development of Compassionate Communities and Compassionate Cities.
**Technical Criteria for the Search**
Studies reporting original data. Qualitative, quantitative and mixed methods studies. Articles written in English and Spanish. Date 2000–2018.
**Types of Studies**
Descriptive studies: describes the characteristics of the population or phenomenon that is being studied. Retrospective and/or prospective studies: results and data analysis have taken place in the past and/or are reported after a period of time has elapsed, respectively. Comparative Studies: examine, compare and contrast subjects or ideas. Evaluation studies: Quantitative methods like surveys, questionnaires and polls and qualitative methods improve decision making. Intervention studies: participants receive some kind of intervention, such as a new medicine, in order to evaluate it. Systematic reviews: systematic methods to collect secondary data, critically appraise research studies, and synthesize findings qualitatively or quantitatively. Meta-analysis: combining data from multiple studies. Case study: is a research strategy and an empirical inquiry that investigates a phenomenon within its real-life context. Ethnographic study: qualitative method where researchers completely immerse themselves in the lives, culture, or situation they are studying. Grounded theory study: a systematic methodology in the social sciences involving the construction of theories through methodical gathering and analysis of data.

**Table 3 ijerph-17-06271-t003:** Registration form.

Number:
Title:
First Author Year:
Reference:
**Document (select X)**
National strategies, Framework Programs, Strategic Plans and Reports,
Consensus documents of associations, institutions, groups of professionals, scientific societies, expert panels, etc. related to the theme.
Reports and studies evaluating the results of the development of Compassionate Communities and Compassionate Cities.
Resolutions and political reports,
Books, White papers,
Tools and guides for program evaluation and measurement of indicators,
Scientific articles,
Grey literature: theses, books, book chapters.
**Type of Study (select X)**
Descriptive, retrospective and / or prospective studies
Comparative Studies
Evaluation studies
Intervention studies
Systematic reviews
Meta-analysis
Case study, ethnographic study, grounded theory study.
**Risk of Bias Assessment (Select X)**
Discuss articles about compassionate communities at the end of life?
Do they express Compassionate Communities and cities development models?
Do they expose development phases or standards?
Are development experiences and results discussed?
Are these pilot studies?
Is the development method described?
Do they describe tools and resources to develop Compassionate Communities and cities?
Is it development from public policies in palliative care?
**Accepted (Select X)**
Articles or other types of documents incomplete or in the process of being drawn up,
Documents that do not include information related to the object of the search,
Documents in other languages than English and Spanish.
**Information Contained (Select X)**
Models of community development and compassionate cities at the end of life.
Evaluation models of Compassionate Communities and Compassionate Cities.
Indicators, standards, or data that allow evaluating the organizations and resources.
Tools and information systems that allow the evaluation of Compassionate Communities and Compassionate Cities.
National regulations and public health models that establish the strategic lines for the development of communities and compassionate cities.
**Primary Results**
Dimension (What is the scope? coverage?)
Objective (What do you want to achieve at city, organization, country level?)
Development model (How do they do it?)
Degree of development (what stage are they in?)
Standards (Are there goals to be achieved? What are the standards for achieving this?)
Tools and resources.
Models of evaluation, reporting and analysis of results.
**Narrative Summary of the Main Results**

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
