# Peer review of "Implementation Models of Compassionate Communities and Compassionate Cities at the End of Life: A Systematic Review"

_ijerph, 2020, doi:10.3390/ijerph17176271_

Round 1
Reviewer 1 Report
Silvia Librada-Flores, María Nabal-Vicuña, Diana Forero-Vega, Ingrid Muñoz-Mayorga & María Dolores Guerra-Martín. Implementation models of Compassionate Communities and Compassionate Cities at the end of life: A systematic review.
This study is important. The analyses is necessary in order to adjust the trajectory of emerging Compassionate Communities/Cities at the end of life.
However the review lacks decisive analyses and as a result meanders around the details fo the studies rather than their outcomes. Although there is also an appreciation of the importance for details, especially in the the methodologies and approach, the paper lacks determination of the efficacy of these different models. Although the conclusion does bring the paper together, the conclusion is disjointed and seems more like bullet points rather than a narrative. What you have to say is important and what service providers want to hear is what works, why it works, and what can be improved.
There are small typographical errrors throughout.
Specific questions relate to:
388 Furthermore, satisfaction at end of life care can be modified by the grief.
How? What type of grief? What extent modified?
There is some repetition that is due to a lack of tight editing, to highlight emerging themes and then to develop your argument on the basis of these themes. Too often the narrative reverts back to the details rather than the outcomes.
284 Crowther et al. [36] published a study conducted to measure the effect of compassion,
285 humanity and kindness on patients with dementia during their last year of life. This is a qualitative
286 analysis based on interviews with 40 caregivers in their own homes or places of work, recorded and 287 transcribed.
and then again:
399 Crowther et al. [36]. They conducted a qualitative study on 40 informal carers of people with
400 dementia where the experiences of compassionate intervention programs were evaluated by a
401 qualitative study.
At this later citation, the paragraph concludes without giving the outcome of the study. The reader is left wondering what is so important.
Author Response
We would like to thank the reviewers for the Deep revision. All his comments help us a lot to clarify many aspects and improve our manuscript.
Comments to the reviewers 1 and 2.
• We consider to eliminate the paragraph 388 “Furthermore, satisfaction at end of life care can be modified by the grief” and the repetition of reference 36 by Crowther et al in discussion section to not repeat the findings from the Results section.
• Minor tips have been checked.
Reviewer 2 Report
Dear Authors,
Thank you for the opportunity to review your manuscript. I offer the following recommendations for your consideration:
- The review is well-described and well-written.
- Contextualize the findings in the Discussion section with respect to the current literature. Use this space to not repeat the findings from the Results section, rather describe how the findings fit into the existing evidence. Also, use this space to discuss what an evaluation system of CCC for end of life care might look like - what should the components of such a system include and provide a rationale.
- There are minor typos - check the spelling of "systematic" and it should be "Theme" on line 383.
Author Response

(The authors gave the same response as above.)
